# IgG^+^ Extracellular Vesicles Measure Therapeutic Response in Advanced Pancreatic Cancer

**DOI:** 10.3390/cells11182800

**Published:** 2022-09-08

**Authors:** Nuno Couto, Julia Elzanowska, Joana Maia, Silvia Batista, Catarina Esteves Pereira, Hans Christian Beck, Ana Sofia Carvalho, Maria Carolina Strano Moraes, Carlos Carvalho, Manuela Oliveira, Rune Matthiesen, Bruno Costa-Silva

**Affiliations:** 1Champalimaud Physiology and Cancer Programme, Champalimaud Foundation, 1400-038 Lisbon, Portugal; 2Digestive Unit, Champalimaud Clinical Centre, Champalimaud Foundation, 1400-038 Lisbon, Portugal; 3Centre for Clinical Proteomics, Department of Clinical Biochemistry and Pharmacology, Odense University Hospital, Sdr. Boulevard 29, DK-5000 Odense, Denmark; 4Computational and Experimental Biology Group, iNOVA4Health, NOVA MedicalSchool|Faculdade de Ciências Médicas, NMS|FCM, Universidade Nova de Lisboa, 1150-199 Lisbon, Portugal; 5Department of Mathematics and CIMA-Center for Research on Mathematics and Its Applications, University of Évora, 7004-516 Evora, Portugal

**Keywords:** pancreatic cancer, extracellular vesicles, liquid biopsy, biomarker, IgG

## Abstract

(1) Background: Pancreatic ductal adenocarcinoma (PDAC) is expected to be the second-leading cause of cancer deaths by 2030. Imaging techniques are the standard for monitoring the therapy response in PDAC, but these techniques have considerable limits, including delayed disease progression detection and difficulty in distinguishing benign from malignant lesions. Extracellular vesicle (EV) liquid biopsy is an emerging diagnosis modality. Nonetheless, the majority of research for EV-based diagnosis relies on point analyses of EVs at specified times, while longitudinal EV population studies before and during therapeutic interventions remain largely unexplored. (2) Methods: We analyzed plasma EV protein composition at diagnosis and throughout PDAC therapy. (3) Results: We found that IgG is linked with the diagnosis of PDAC and the patient’s response to therapy, and that the IgG+ EV population increases with disease progression and reduces with treatment response. Importantly, this covers PDAC patients devoid of the standard PDAC seric marker CA19.9 expression. We also observed that IgG is bound to EVs via the tumor antigen MAGE B1, and that this is independent of the patient’s inflammatory condition and IgG seric levels. (4) Conclusions: We here propose that a population analysis of IgG+ EVs in PDAC plasma represents a novel method to supplement the monitoring of the PDAC treatment response.

## 1. Introduction

Pancreatic ductal adenocarcinoma (PDAC) was the seventh cause of cancer-related fatalities globally in 2020 [1]. As reported by the Surveillance, Epidemiology, and End Results Program (SEER) [2], the incidence of PDAC has increased over the past seven years [3], being projected to be the second-leading cause of cancer-related deaths by 2030 [4]. The majority of metastatic patients continue to have median survival outcomes of less than one year [5,6], with less than 30% of patients eligible for second-line chemotherapy [7]. 

Imaging techniques based on the response evaluation criteria in solid tumors (RECIST) criteria [8], which evaluate the number and size of tumor lesions over the course of treatment, remain the gold standards for determining treatment response during chemotherapy. However, these techniques have several drawbacks, including a lack of precision in detecting small tumors and an inability to differentiate between benign inflammatory and malignant lesions. This also includes the necessity to arbitrarily target lesions to evaluate the evolution of the disease in accordance with the treatment, and the usual delay between the imagiological response and the real-time detection of differences in tumor dimensions [9]. In addition, the evaluation solely based on dimensions does not consider changes in tumor attenuation, nor does it discriminate viable cells from non-viable ones, thus complicating the measurement of the treatment response [10]. 

Serological markers, such as carbohydrate antigen 19.9 (CA19.9), are widely utilized to supplement the ability of RECIST to assess PDAC treatment response [11]. When elevated at the start of treatment, CA19.9 demonstrates dependable outcomes in conjunction with imaging evaluation [12]. However, the CA19.9 evaluation in PDAC patients has limitations, since it is not expressed in 5–20% of patients [12,13]. In addition, false positives are common, especially in patients with obstructive lesions of the biliary tract, which may affect up to 70% of PDAC patients [14]. 

Due to the aforementioned constraints, it is of the utmost need to develop additional techniques for assessing the therapy response of PDAC. Studies using potential new response indicators, including circulating tumor cells and circulating tumor DNA, have previously been conducted to uncover new readouts [15]. This also includes our recent findings that plasma extracellular vesicles (EVs) can detect incidence [16], predict progression [17], and locate [18] metastases in PDAC patients. However, the majority of the current research lacks longitudinal information regarding the response to therapy and clinical follow-up.

EVs are nanovesicles that contain all the active biomolecules, including nucleic acids, lipids, and proteins [19]. EVs are critical mediators of cell-to-cell communication under physiological and pathological settings [20]. Studies using EVs as cancer biomarkers usually focus on specific disease timepoints, disregarding potential changes in the molecular composition and population dynamics of EVs during the course of the disease, and investigating the utility of EVs exclusively for prognosis or diagnosis [21]. Moreover, studies of EV biomarkers for follow up of metastatic PDAC patients have been limited by challenges involving their dismal prognosis, which complicates patient recruitment for longer longitudinal studies. 

In this study, we characterized EVs obtained from metastatic PDAC patients who were monitored throughout their treatment and the evolution of the disease. We identified a population of plasma EVs bound to IgG (IgG+ EVs) that is connected with the diagnosis of PDAC and the treatment response of PDAC patients (including those without CA19.9 expression). We demonstrated that longitudinal examination of IgG+ plasma EVs may represent a new tool for enhancing the efficacy of chemotherapeutic treatments in patients with PDAC by improving the detection of therapy response and resistance.

## 2. Materials and Methods

### 2.1. Patients and Controls

Patients were recruited between May 2016 and October 2019. Patients were eligible to participate in this study if they had a confirmed histological or cytological diagnosis of metastatic PDAC, according to the 8th edition of AJCC Cancer Staging [22], and were able to receive chemotherapy. Samples from healthy individuals were collected from voluntary donors at the Champalimaud Clinical Center and the Champalimaud Research (Champalimaud Foundation, Lisbon, Portugal). Exclusion criteria were previous diagnosis of other tumors or inflammatory diseases. Written informed consent was obtained from both patients and control individuals, and samples were deidentified for confidentiality. All the procedures were conducted in accordance with the Helsinki Declaration and its amendments, with the approval of the Ethical Committees of Nova Medical School and of the Champalimaud Foundation.

Treatment proposals were presented to the multidisciplinary tumor board. Patients were treated with a sequence of chemotherapy in the following approved regimens: association of 5-fluorouracil, irinotecan, and oxaliplatin (FOLFIRINOX); association of gemcitabine and nab-paclitaxel; association of 5-fluorouracil and liposomal irinotecan; or single gemcitabine. The decision regarding the choice of the chemotherapy regimen was at the discretion of the treating oncologist. Demographic and clinical information from the patients, including neutrophil-to-lymphocyte ratio (NLR) and C-reactive protein (CRP) levels, was collected. Treatment response was classified according to the imaging response based on RECIST v1.1 criteria [8]. Timepoints were selected as follows: timepoint I (pretreatment) was collected before the beginning of a new chemotherapy regimen, and timepoint II (posttreatment) at the time of either the best imaging response (if the tumor responded to chemotherapy) or the worst (if the tumor did not respond to chemotherapy). The same patient can be a responder at one point of treatment and a nonresponder at another. The first samples of one patient that met these criteria were selected for the study until the calculated minimum number of total required samples was obtained. 

A total of 5 of the 19 patients who participated of our study were used exclusively for the proteomic analysis of EVs, as there were no samples left for the subsequent experiments. The remaining 14 patients, with a total of 155 plasma samples, were studied by vesicle flow cytometry. Figure 1 provides information on the metastatic profile, treatment regimens, and sample collection times. For the control group, we applied the same exclusion criteria (previous oncological diagnosis or previous inflammatory diseases). The group included 21 individuals (ten males and eleven females) with a mean age of 47 years (minimum of 25 years; maximum of 85 years). 

### 2.2. Purification and Characterization of EVs from Plasma

Blood samples from patients and healthy donors were collected in 9 mL Vacuette NV EDTA K3 tubes and centrifuged twice at 10 °C (500 g for 10 min and 3000 g for 20 min). Prior to analysis, plasma samples were aliquoted and stored at −80 °C. A protocol we previously described involving sequential ultracentrifugation combined with a sucrose cushion was used to purify the EVs [23]. 

All EV samples were analyzed by nanoparticle tracking analysis (NTA) using a NanoSight NS300, equipped with a red laser (638 nm), to determine particle concentration and size distribution (Malvern Panalytical, Malvern, UK). Samples were pre-diluted in filtered PBS to achieve a concentration within the range for optimal NTA analysis. Video acquisitions were performed at 25 °C using a camera level of 16, and a threshold between 4 and 6. Five videos of 30 s with 10–50 particles per frame were captured per sample. The total protein content of the EV samples was determined using the PierceTM BCA Protein Assay Kit (Thermo Fisher Scientific, Waltham, MA, USA).

### 2.3. Characterization of EV Protein Composition by Mass Spectrometry (MS)

For evaluation by MS, plasma-derived EV samples from PDAC patients were used. Four patients who responded, and four patients who displayed disease progression were selected. For each patient, samples were collected at diagnosis and after treatment response evaluation, totaling 16 samples. In parallel, 5 plasma-derived EV samples from healthy controls were also compared to 8 PDAC patients at diagnosis. 

The EV solution containing sodium dodecyl sulfate (SDS) and dithiothreitol (DTT) was loaded onto filtering columns and washed exhaustively with 8 M urea in HEPES buffer [24]. The proteins were reduced with DTT and alkylated with IAA. Protein digestion was performed by overnight digestion with trypsin sequencing grade (Promega).

Peptides samples were analyzed by nano-LC-MSMS (Dionex RSLCnano 3000) coupled to an Exploris 480 Orbitrap mass spectrometer (Thermo Scientific, Hemel Hempstead, UK) virtually, as previously described [25]. Briefly, samples were loaded (flow rate 5 µL per minute for 6 min) onto a custom-made fused capillary pre-column (2 cm length, 360 µm OD, 75 µm ID) packed with ReproSil Pur C18 5.0 µm resin (Dr. Maish, Ammerbuch-Entringen, Germany), followed by the separation on a custom-made fused capillary column (25 cm length, 360 µm outer diameter, 75 µm inner diameter) packed with ReproSil Pur C18 1.9-µm resin (Dr. Maish, Ammerbuch-Entringen, Germany) using a flow rate of 250 nL per minute. The gradient was from 89% A (0.1% formic acid) to 32% B (0.1% formic acid in 80% acetonitrile) over 56 min. The mass spectra were acquired in positive ion mode by applying an automatic data-dependent switch between one Orbitrap survey MS scan in the mass range of 350–1200 m/z, followed by a higher-energy collision dissociation (HCD) fragmentation and Orbitrap detection of fragment ions, with a cycle time of 2 s between each master scan. MS and MSMS settings: the maximum injection times were set to “Auto,” the normalized collision energy was 30%, the ion selection threshold for MSMS analysis was 10,000 counts, and the dynamic exclusion of sequenced ions was set to 30 s.

Data obtained from 46 LC-MS runs of 21 subjects and 32 LC-MS runs of IgG bound proteins were searched using VEMS [26,27] and MaxQuant [28]. A standard proteome database from UniProt (3AUP000005640), in which common contaminants were included, was also searched. Trypsin cleavage, allowing a maximum of 4 missed cleavages, was used. Carbamidomethyl cysteine was included as a fixed modification. Methionine oxidation and N-terminal protein acetylation were included as variable modifications; 5 ppm mass accuracy was specified for precursor ions and 0.01 m/z for fragment ions. The FDR for protein identification was set at 1% for peptide and protein identifications. No restriction was applied to the minimal peptide length for the VEMS search. Identified proteins were divided into evidence groups, as defined in [27]. Functional pathway analysis was performed with STRING (string-db.org). 

### 2.4. Western Blotting 

Western blotting was used to assess the presence of EV and non-EV protein markers. Equal protein amounts of EV samples were mixed with 4X Laemmli buffer (Bio-Rad, Herculaes, CA, USA), denatured for 5 min at 95 °C, and loaded onto 4–20% Mini-PROTEAN TGX Stain-Free Protein Gels (Bio-Rad). SDS-PAGE was run for 1.5 h at 90 V, and then proteins were transferred to nitrocellulose membranes (Cytiva) at 100 V for 1 h. Membranes were blocked with LI-COR Intercept Blocking Buffer (LI-COR Biosciences) for 1 h at RT. Blocked membranes were incubated overnight at 4°C with primary antibodies diluted in LI-COR blocking buffer with 0.1% Tween-20. Membranes were washed with TBS-T (TBS with 0.1% Tween-20) 3 times for 5 min, and then incubated with secondary antibodies for 1 h at RT. Incubation was followed by 3 additional washes with TBS-T, 5 min each. Blots were imaged using the Odyssey Infrared Imaging System (LI-COR Biosciences). The detailed list of primary and secondary antibodies used is provided in Appendix A.

### 2.5. Analysis of IgG+ EV Population by Vesicle Flow Cytometry

Flow cytometry analysis of plasma EVs was performed as described by our group [29]. A volume of plasma containing 2 × 10^9^ particles was used for staining with 0.5 μL of anti-IgG (Abbexa abx142503, Houston, TX, USA) in PBS, in a final volume of 40 μL, and incubated for 1 h at 37 °C. The antibody-stained sample was then incubated with Carboxyfluorescein Diacetate Succinimidyl Ester (CFSE—Thermo Fisher Scientific LTI C34554, Waltham, MA, USA) to a final concentration of 25.6 μM, for 90 min at 37 °C. For the removal of unbound CFSE and antibody, Size Exclusion Chromatography (SEC) columns (iZON qEV original columns SP1, UK) were used. EV-enriched fractions #7, #8, and #9 were then pooled (total of 1500 μL) and retrieved for analysis with the flow cytometer Apogee A60-Micro-Plus (Apogee Flow Systems, Hertfordshire, UK), configured as described in Appendix A. For all subsequent analyses, quadrant thresholds were established with unstained and single-stained extracellular vesicles (with CFSE or with anti-IgG) (Appendix A).

Internal controls across assays were performed as previously described [29]. The acquired data was exported and analyzed with FlowJo software v10.4.2 (FlowJo LLC, Ashland, OR, USA).

### 2.6. Total Plasma Immunoglobulin G Quantification

For the quantification of the plasma IgG, samples were processed using the nephelometry method (BNProSpec—Synlab, Lisbon, Portugal). The reference values for healthy controls were 700–1 600 mg/dL.

### 2.7. Identification of EV Surface Proteins Associated with IgG 

To identify which surface proteins of EVs bind to IgG, 200 μg of EVs isolated from 8 PDAC patients and 8 healthy controls were first biotinylated (EZ-Link™ Sulfo-NHS-SS-Biotin, Thermo Scientific, 21331) and then lysed using 0.5% NP-40 lysis buffer (150 mM NaCl, 10 mM Tris-HCl pH 7.5, 0.5 mM EDTA, 0.5% NP-40). Next, the biotinylated surface proteins of EVs were collected using streptavidin magnetic beads [30] (Dynabeads™ MyOne™ Streptavidin C, Invitrogen, 65001), and then detached from the beads (deionized water, 70 °C)[31]. After separation, surface EV proteins that were associated with IgG were co-immunoprecipitated using magnetic beads (Dynabeads™ Protein G for Immunoprecipitation, Invitrogen, 10003D), conjugated with anti-human IgG antibody (Goat anti-Human IgG F(ab’)2 Secondary Antibody, Invitrogen, 31122), then eluted and analyzed by MS. 

### 2.8. Statistical Analysis

The sample size was based on previous liquid biopsy studies [32]. Our study involves 155 observations from 30 different situations (15 responders x 15 nonresponders). Experiments were not randomized. The researchers were blinded to allocation during the experiments and the outcome assessment. The response evaluation to the treatment was previously performed by a different researcher. Error bars in graphical data represent means ± standard errors of the means (SEM). Normality and homogeneity of variances from the analyzed variables were tested with the Shapiro–Wilk test and the Bartlett or Levene tests, respectively. If data were parametric, Student’s t tests (two populations) were used. If data were not parametric, Wilcoxon or Mann–Whitney tests were performed. For ROC analysis, values were obtained from the division of IgG+EVs and CA19.9 readings after treatment by the values obtained before treatment in each studied point, as previously described [11]. The statistical packages used was R v.4.0.2. For all evaluations, a *p*-value under 0.05 was considered statistically significant, and the null hypothesis was rejected. Graphical design was performed with the GraphPad Prism software (GraphPad software).

## 3. Results

### 3.1. Identification of Possible EV Markers for PDAC Diagnosis and Therapeutic Response 

The size distribution and concentration of plasma EVs isolated from patients and healthy controls were characterized. Proteins frequently present or absent in small EVs were measured in our samples (Appendix A). We next performed MS analysis of plasma EV samples from 5 healthy controls and 16 samples from 8 PDAC patients, both at the time of diagnosis and after treatment. A total 4 of these 8 patients were considered chemotherapy responders, as tumor shrinkage was observed between the diagnosis and treatment timepoints. In contrast, based on the observed imaging progression of the disease between the 2 time points, the remaining 4 patients corresponded to nonresponders to chemotherapy. For the MS analysis, the same amounts of protein (20 μg) and concentrations (0.5 μg/μL) were utilized. 

Protein expression analysis revealed that 102 distinct proteins exhibited statistically significant differences between PDAC patients and healthy controls, 59 of which were upregulated in PDAC patients. Of these, we identified the presence of multiple IgG fragments (Figure 2A, Appendix A). In fact, the functional analysis of proteins significantly upregulated or downregulated in EVs from PDAC patients (responders and nonresponders) and as compared to healthy controls, revealed enrichment in proteins associated with humoral immune response and complement activation, among others (Figure 3).

We were also interested in identifying treatment response indicators. By comparing EVs isolated from PDAC patients who responded to therapy with those who did not, we identified 43 proteins that exhibited statistically significant differences between responders and nonresponders, with 24 of these upregulated in nonresponders (Figure 2B, Appendix A). We found that 16 of the upregulated proteins in patients who did not respond to chemotherapy were IgG fragments. As a result, we chose to further investigate whether the presence of IgG in populations of plasma EVs is indeed associated with PDAC diagnosis and therapeutic response.

### 3.2. Evaluation of IgG+ EVs as Possible Markers of PDAC Treatment Response 

In accordance with the MS findings, vesicle flow cytometry revealed that metastatic PDAC patients have a larger population of IgG+ EVs compared to healthy control donors (Figure 4A), suggesting that IgG+ EVs may be a useful diagnostic marker for advanced PDAC disease. Next, we examined whether this IgG+ EV population could be used to determine whether a patient is responding to chemotherapy or not. 

At and after diagnosis, prospective clinical information, CT data, serum levels of CA19.9, hemograms, and serial whole-blood results were collected from patients with stage IV PDAC. To be considered suitable in clinical practice for the care of metastatic PDAC patients, a new marker should: (a) be consistent, independent of the treatment of choice; (b) be able to identify differences in patients with and without CA19.9 expression; and (c) be able to predict treatment response in comparison to the imagiological evaluation, in order to reinforce the maintenance of applied treatment or to suspend futile treatments early [33,34]. Keeping this in mind, we analyzed the proportion of IgG+ EVs in the plasma of multiple patients who had received at least two lines of chemotherapy (FOLFIRINOX—first line; Gemcitabine/nabpaclitaxel—second line). In contrast to the increase in IgG+ EV populations in the plasma of PDAC patients during tumor progression (nonresponse), the monthly evaluation of IgG+ EV populations in the plasma of PDAC patients (Figure 4B) revealed a decrease in IgG+ EVs upon imagiological response.

To determine whether IgG+ EVs meet the criteria for independence from CA19.9 expression, we performed the same monthly IgG+ EVs analysis on two patients without CA19.9 expression (patients 8 and 63). The same pattern was observed as for patients with CA19.9 expression (patients 15, 23, 34 and 49), suggesting that the evaluation of IgG+ EVs as a marker of response to therapy may be applicable to all metastatic PDAC patients (Figure 4B). In addition, we compared the measurement of IgG+ EVs to that of CA19.9, the gold standard serological marker in PDAC [12]. We observed a significant downregulation of CA19.9 in patients who responded to therapy (Appendix A) while no significant upregulation of CA19.9 was observed in nonresponders (Appendix A). 

In addition to the individual longitudinal analysis of patients, we grouped chemotherapy response and nonresponse timepoints to investigate the population of IgG+ EVs in responders and nonresponders. As shown in Figure 4C, we observed a significant decrease in the IgG+ EV population in responders following treatment. In contrast, the proportion of IgG+ EVs significantly increased in patients who did not respond to chemotherapy (Figure 4D). The evaluation of sensitivity and specificity of IgG+EV for response evaluation of patients with metastatic PDAC showed an AUC of 0.8311 (95% confidence interval 0.6788 to 0.9834, *p* = 0.0020) (Figure 4E), compared to an AUC of 0.9911 (95% confidence interval 0.9679 to 1.00, *p* < 0.001) for the same evaluation for CA19.9 in our population (Appendix A).

Our findings suggest that levels of IgG+ EVs are associated with imaging evaluations of clinical response to treatment, suggesting that IgG+ EVs could be used as a readout in PDAC clinical settings.

### 3.3. Evaluation of the potential role of Plasmatic IgG Levels and Inflammation Status in IgG+ EV Proportion 

The fluctuations in the plasmatic proportion of IgG+ EVs could, in principle, be a direct result of variations in plasma IgG [35] levels. To test this hypothesis, the total amount of plasmatic IgG was measured in the same set of patients. We found no significant differences in plasmatic IgG levels during treatment response or disease progression in patients with PDAC (Appendix A). We found no correlation between the percentage of IgG+ EVs and total plasmatic IgG levels, indicating that the variation in IgG+ EVs is not due to fluctuations in plasmatic IgG levels (Appendix A). 

Alternately, IgG+ EVs fluctuations could derive, at least in part, from alterations in the inflammatory status of immune cells during the progression of PDAC. The NLR is a clinical marker of inflammation calculated as the quotient of the absolute neutrophil and lymphocyte counts [36]. In light of this, we found no differences between the groups (responders and nonresponders) and no correlation between levels of IgG+ EVs and NLR. In addition, we found no correlation between the levels the inflammatory marker CRP [37] and the levels of IgG+ EVs (Appendix A), further suggesting that in the context of our study, the inflammatory status has no effect on IgG+ EV levels. 

### 3.4. Identification of IgG-Associated Proteins on the Surface of PDAC Patient EVs 

Next, we investigated potential differences in the molecular mechanisms of IgG transport by plasma EVs between healthy controls and PDAC patients. To test this, we devised a method for characterizing protein interactions at the surface of the EVs. Briefly, biotinylated surface proteins of lysed EVs were isolated by streptavidin binding, followed by co-immunoprecipitation with an antigen-specific antibody. Here, we precipitated IgG from the mixture and searched for EV surface proteins bound to IgG. 

Several IgG fragments were detected in both healthy controls and PDAC patients. In addition, soluble proteins (alpha-2-macroglobulin), cell surface receptors (i.e., glutamine receptor), cytoskeletal proteins (i.e., keratin), and cytoplasmic proteins were detected in both groups (e.g., Chloride intracellular channel protein 4, Golgi integral membrane protein 4, Chondroitin sulfate synthase 3, and Protein Argonaute 2). IgG was bound to melanoma associated antigen B1 (MAGE B1) in the EVs of PDAC patients, in addition to IgG fragments and albumin. MAGE B1 is a well-known PDAC antigen [38], suggesting that the population of IgG+ EVs described here is the consequence of an interaction between tumoral antigens on the surface of EVs released by tumor cells and IgG in circulation (Table 1).

## 4. Discussion

Currently, the assessment of treatment response in patients with metastatic PDAC relies mainly on CA19.9 and imaging evaluations, both of which have limitations. A total of 5–20% of PDAC patients lack CA19.9, and false-positive elevations are associated with biliary tree inflammation or infections [12,13]. On the other hand, imaging evaluation cannot detect small lesions and is not an immediate indicator of the tumor’s status, as there is a delay between the progression of the disease and its imagiological identification [8,10]. By characterizing a population of plasma extracellular vesicles (IgG+ EVs) that correlates with the treatment response status of metastatic PDAC patients, we have described a potentially new tool to complement treatment response evaluation.

EVs have the potential to carry biomarkers for liquid biopsies, in both oncologic and non-oncologic diseases [39], due to their abundance in body fluids. EVs produced by tumor cells can populate and alter the composition of biofluids. As a result, the identification of molecular modifications in EVs associated with tumor profile has been utilized to identify putative cancer biomarkers[19,40], including the diagnosis of PDAC [41]. The majority of these studies, however, have focused on the differential expression of specific nucleic acids and proteins in bulk EV samples for diagnosis or prognosis, using single collections from each patient [42]. In cancer patients, longitudinal studies like ours evaluating the response to chemotherapy in a single individual are still scarce [43,44]. To our knowledge, this is the first study to examine the dynamics of circulating EV populations in patients with metastatic PDAC.

The analysis of EV bulks is an effective method for identifying molecules of interest (e.g., proteins, lipids, and RNA) in EV liquid biopsies. In fact, we relied on MS analysis of EVs in bulk to identify IgG as a possible EV marker of therapeutic response in PDAC patients. However, failure to distinguish between EV populations may obscure real differences between experimental groups. Moreover, the implementation of EV biomarkers in clinical practice is hindered by the laborious and time-consuming isolation and analysis protocols commonly employed for EVs. In an effort to characterize populations of extracellular vesicles (EVs) using a method that permits rapid analysis of markers in EV samples, we utilized a vesicle flow cytometry protocol described by our group [29]. By not requiring EV isolation prior to analysis, the processing time is reduced from >24 h to 4 h. Therefore, the use of vesicle flow cytometry has the potential to facilitate the clinical evaluation of IgG+ EVs.

Although the main objective of our work was to identify a novel biomarker to monitor the therapy response of diagnosed metastatic PDAC patients, instead of identifying a new marker to diagnose PDAC, we found elevated levels of IgG+ EVs in PDAC patients compared to healthy controls, reinforcing previous descriptions of the potential application of this marker for PDAC diagnosis [45]. In our study, we found that levels of IgG+ EVs did not correlate with validated clinical markers for inflammation (NLR and CRP), suggesting that this biomarker is not affected by the inflammatory background of the patients. However, as the recruitment criteria for our study excluded individuals with inflammatory conditions, further study will be necessary to address the potential impact of an inflammatory background on the specificity of IgG+ EVs as a diagnostic marker for PDAC.

In our longitudinal studies, we were able to observe the dynamics of IgG+ EV populations within each PDAC patient and their association with the evaluation of chemotherapy response. We also demonstrated that the study of IgG+ EV populations may be utilized in the follow-up of PDAC patients, including those who lack CA19.9 expression. Due to the absence of this established marker, these patients rely solely on imaging evaluations to determine their clinical response to chemotherapy; therefore, a new reliable marker would represent a substantial improvement in their care.

During the characterization of proteins bound to IgG in circulating EVs, a number of proteins were identified in both healthy donors and PDAC patients. This suggests that at least some IgGs bind to EVs via protein–protein interactions that may occur after EVs are secreted by cells. Consequently, circulating IgG and IgG ligands in EVs may interact in the extracellular environment of both cancer patients and healthy individuals [46]. This is further supported by previous studies on autoimmune diseases, showing that immunoglobulins can bind to circulating EVs and form immune complexes that contribute to disease pathology [47,48,49,50]. In fact, the composition of EVs is not merely a result of their intracellular biogenesis, since their surface is highly interactive with proteins present at the extracellular milieu. The interaction of EVs with secreted proteins has been shown to modulate their immune recognition, mobility, uptake, and signaling capabilities [51]. Remarkably, many proteins that frequently display quantitative changes in cancer patients, such as cytokines/chemokines [52], extracellular matrix proteins [53], coagulation factors [54], complement factors [55], immunoglobulins [56], and albumin [57], can interact with EVs after their release and change their composition [51].

The MS analysis of proteins bound to IgG focused on EV surface proteins, as only these proteins were biotinylated during the immunoprecipitation protocol. Moreover, as vesicle flow cytometry studies were performed with intact EVs, all IgG^+^ signals originated from IgG bound at the external face of the EV surface. This agrees with the conventional mechanism of IgG binding to the surface of cells, which depends on the binding of IgG to a cell membrane protein on B cells, the Igα/Igβ heterodimer (CD79α/CD79β) [58,59]. Therefore, we consider that, as to CD79α/CD79β, at least part of the IgG anchorage on EVs relies on binding to surface proteins of EVs, such as MAGE B1. Although MAGE B1 was identified as an EV surface ligand of IgG in 8 different PDAC patients, further studies which include methods other than MS will be necessary to validate this finding in larger cohorts of PDAC patients. Importantly, although immunoglobulins were traditionally thought to be exclusively produced by B-lineage cells, recent studies have shown that these molecules can also be produced by a large diversity of tumor types [21], including PDAC [35,60,61]. Therefore, although the association with MAGE B1 suggests that IgG binding to circulating EVs may be the result of post-secretion interactions with tumor neoantigens present in tumor EVs, we cannot deny that at least part of the circulating IgG+EVs associated with metastatic PDAC burden could potentially be packed and secreted at the surface of EVs directly by the PDAC cells.

MAGE B1 is a tumor antigen found in a variety of tumor types, including melanoma and tumors of epithelial origin, such as breast, colorectal carcinoma, lung, and pancreatic [38,62,63,64,65,66] tumors. In addition, MAGE is identified as an antigen normally expressed by the placenta and male germ cells in cancerous testes. It is expressed in 47% of pancreatic tumors [67], offering cells that express it a survival advantage [68] and negatively correlating with prognosis and patient survival [67,69]. MAGE B1 was identified as one of the proteins found exclusively in EVs from PDAC patients. In addition, we found that healthy controls had a significantly smaller IgG+ EV population, with the presence of IgG on the surface of these EVs being unrelated to neo-antigens such as MAGE B1 (Table 1). Although the correlation between IgG+ EVs and the response of PDAC patients to chemotherapy is insufficient to conclude that the IgG+ EV population, which varies based on chemotherapy response, is tumor derived, we propose that at least a portion of the IgG+ EVs upregulation during PDAC progression may be due to IgG binding with PDAC EVs expressing MAGE B1. Due to sample limitations, we were not able to further validate this result by additional methods. In spite of that, as the interaction of IgG with MAGE1 and other proteins was verified in all of the 8 different PDAC patients studied in our MS analysis, we are confident that our results are unlikely to be false-positive.

We also found that the proportion of IgG+ EV is independent of the availability of circulating IgG. Alternatively, this may be the result of elevated levels of tumor EV secretion and/or enhanced packaging of tumor antigens (such as MAGE B1) in PDAC EVs. Moreover, it is plausible that this process may result in tumor-directed IgG absorption by tumor EVs and, as a result, may contribute to tumor immune-escape [45] and the chronic inflammatory state observed in metastatic PDAC patients [70]. The binding of IgGs to EVs could also affect the efficacy of targeted therapies (e.g., immunotherapies). Additional research will be required to fully comprehend the aforementioned implications.

## 5. Conclusions

We demonstrate that IgG attaches to the surface of EVs in PDAC patients via an interaction with the tumor antigen MAGE B1, and that this process is independent of IgG plasma levels and the inflammatory status of the patient. For the first time, a longitudinal population analysis of plasma EVs revealed that EVs bound to IgG increase during disease progression and decrease when patients react to chemotherapy. Importantly, IgG+ EVs can detect therapy response in a subset of individuals with PDAC who lack the standard PDAC marker, CA19.9. These findings not only have the potential to expand the current monitoring options for PDAC metastasis, but also represent a promising new tool for identifying effective treatments and indicating alternate treatments, in the event of disease progression. In addition, emerging markers of therapy response in PDAC, such as IgG+ EVs, should assist in separating PDAC patients into new groups, hence aiding the development of tailored, more effective treatments.

## Figures and Tables

**Figure 1 cells-11-02800-f001:**
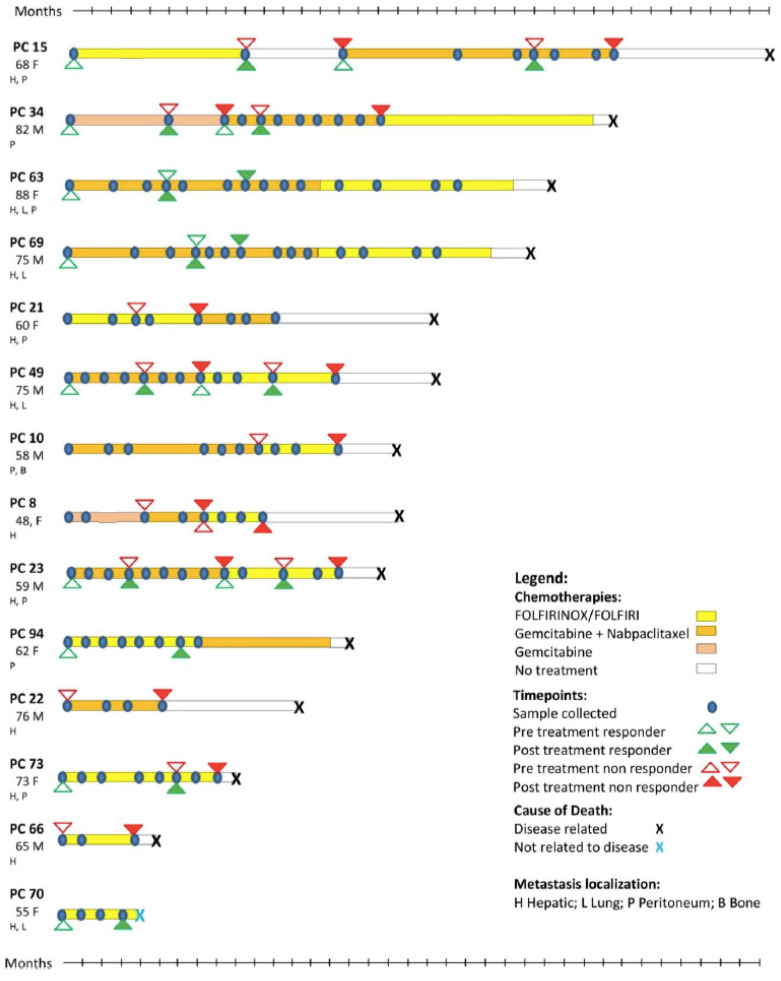
Patients and their respective treatments. Depiction of the metastatic profile, treatment protocols, and sample collections from patients analyzed in our study. PDAC patient (PC) number, age, gender (M,F), and metastatic location(s) are provided.

**Figure 2 cells-11-02800-f002:**
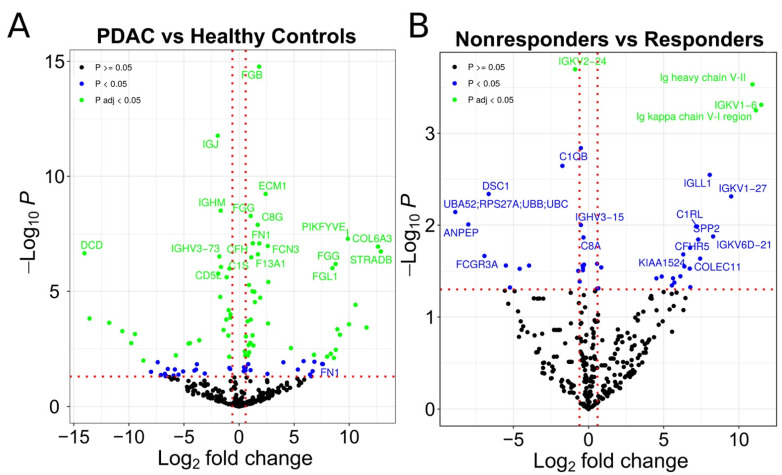
Proteins differentially expressed in EVs. (**A**) Volcano plot representing the identified proteins in MS (comparison between patients (PDAC) and healthy control samples). (**B**) Volcano plot representing the identified proteins in mass spectrometry (comparison between patients that are nonresponders and responders to chemotherapy). The green points represent proteins significantly regulated after correction for multiple testing. The blue points represent proteins significantly regulated without correction for multiple testing. The black points represent proteins with insignificant regulation.

**Figure 3 cells-11-02800-f003:**
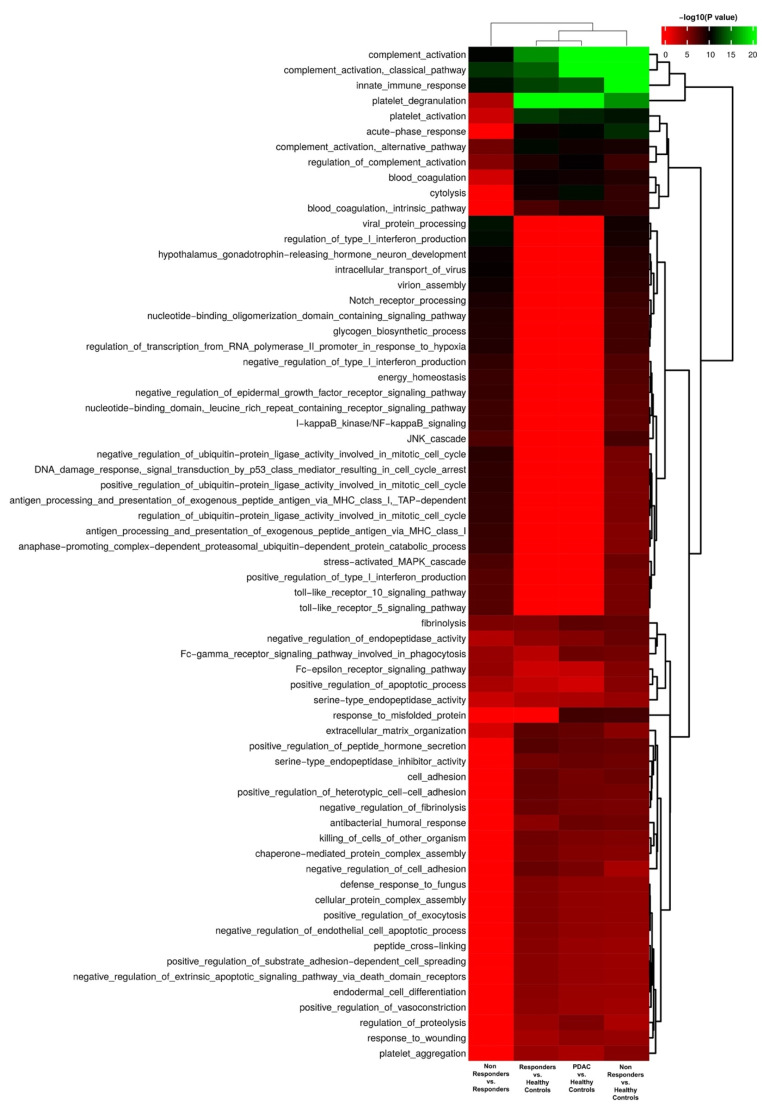
Functional enrichment of significantly regulated EV proteins based on biological processes.

**Figure 4 cells-11-02800-f004:**
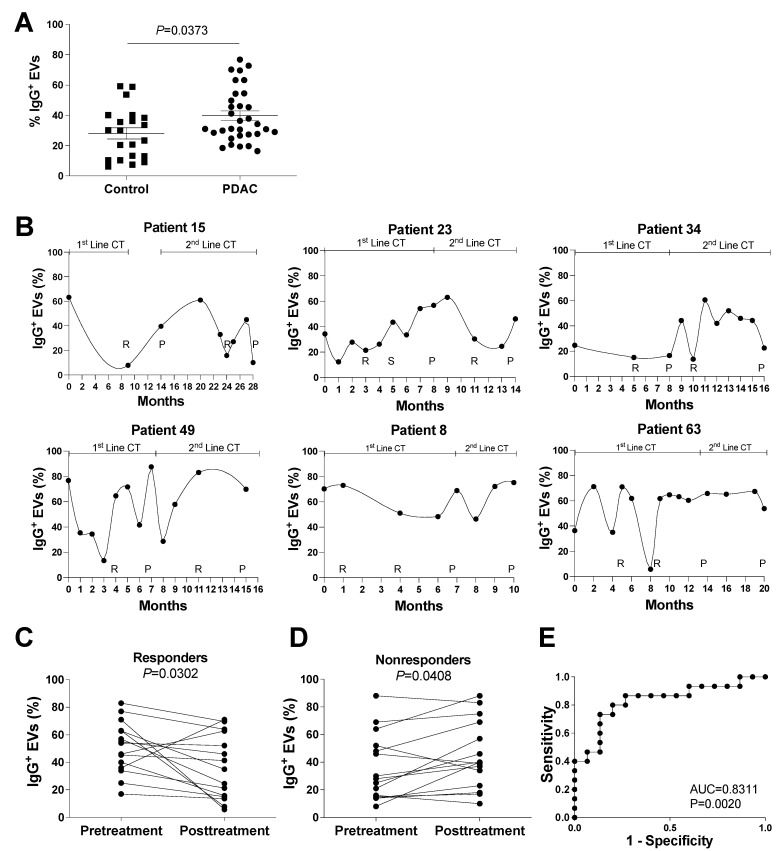
IgG+ EVs in healthy controls and PDAC patients. Vesicle flow cytometry data of plasma samples from PDAC patients and healthy controls, as indicated. (**A**) Comparison of IgG+ EVs between PDAC patients and healthy controls, *p* = 0.0373, by Wilcoxon test. (**B**) Longitudinal evaluation of plasma IgG+ EVs in patients followed during at least 2 lines of chemotherapy (mFOLFIRINOX in the 1st line, and Gemcitabine with Nabpaclitaxel in the 2nd line, as indicated), that express CA19.9 (patients 15, 23, 34, and 49); or do not express CA19.9 (patients 8 and 63). The moments of imagiological evaluation are indicated (R for response, S for stabilization, and P for progression to chemotherapy). (**C**,**D**) Evolution of IgG+ EVs before and after treatment with chemotherapy in 15 situations of response (patients 15, 23, 34, 49, 63, 69, 70, 73, and 94) (**C**) and no response to treatment (patients 8, 10, 15, 21, 22, 23, 34, 49, 66, and 73) (**D**). *p* = 0.0302 (**C**) and *p* = 0.0408 (**D**), by t-test. (**E**) ROC curve for the IgG+ EV to discriminate response in patients with metastatic PDAC.

**Table 1 cells-11-02800-t001:** IgG associated proteins identified by MS.

Cancer Patients	
	Immunoglobulin lambda variable 3–21
	Melanoma-associated antigen B1
	Albumin
	Probable non-functional immunoglobulin heavy variable 3–16
**Healthy Donors +** **Cancer Patients**	
	Keratin, type II cytoskeletal 1
	Chloride intracellular channel protein 4
	Alpha-2-macroglobulin
	Glutamate receptor ionotropic, kainate 3
	Keratin, type II cytoskeletal 6B
	HUMAN Ig kappa chain V-I region Lay
	HUMAN Ig kappa chain V-III region NG9 (Fragment)
	HUMAN Ig kappa chain V-III region POM
	HUMAN Ig kappa chain V-III region CLL
	HUMAN Ig kappa chain V-III region VH (Fragment)
	Immunoglobulin kappa variable 3D-7
	Immunoglobulin kappa variable 3/OR2-268 (non-functional)
	Immunoglobulin kappa constant
	Immunoglobulin kappa variable 3–15
	Probable non-functional immunoglobulin kappa variable 3–7
	Ig lambda chain V-II region NIG-84
	Chondroitin sulfate synthase 3
	Ig lambda chain V-II region BUR
	Keratin, type I cytoskeletal 10
	Protein argonaute-2
	Golgi integral membrane protein 4
**Healthy Donors**	
	S100A9
	Ig kappa chain V-III region Ti
	Uncharacterized protein (Fragment)

## Data Availability

The data presented in this study are available upon request from the corresponding author.

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
