# Peer review of "IgG+ Extracellular Vesicles Measure Therapeutic Response in Advanced Pancreatic Cancer"

_cells, 2022, doi:10.3390/cells11182800_

Round 1

Reviewer 1 Report

In the manuscript “IgG+ extracellular vesicles measure therapeutic response in advanced pancreatic cancer” Couto and colleagues identify IgG+ EVs to be elevated in the plasma of pancreatic cancer patients and investigate their dynamics during chemotherapy. They demonstrate that IgG+ EVs levels correlate with treatment response and therefore represent a promising novel tool for PDAC therapy monitoring. The manuscript is clearly written and the clinical follow-up of the patients is documented in detail. As not many longitudinal studies have been performed for EVs as cancer biomarkers, this study is of high interest and has great potential for clinical translation. However, I would recommend to add some additional information and experiments to support the claims made by the authors:

Major points:

While the authors give information on the PDAC patients used in the study, there is next to no information on the control group. Therefore, I would ask the authors to include information on the cohort of healthy controls used for vesicle flow cytometry (age, gender). Moreover, to confirm the applicability of using IgG+ EVs as method for the detection of advanced PDAC, the authors should include benign controls, e.g. chronic pancreatitis, in particular as it is known that the described marker CA19.9 (and also IgG) is prone to variation under inflammatory conditions. Moreover, a ROC analysis would be helpful to judge the potential of IgG+ EVs as diagnostic marker.

Currently, it is difficult to match the information of imaging response and course of IgG+ EV levels with the exact sample used for the evaluation of treatment response. For each patient the timepoints of sample collection have been indicated in Figure 1. However, not all of the measurements are shown in Figure 4B+C. Why? Moreover, the timelines shown in Figure 1 and Figure 4B+C do not match? The authors state that patients were allocated to the responder or non-responder group based on the timepoint of the best, or worst, imaging response. For example, for patient 15 the best imaging response would be for the first-line treatment in month 3 (Figure 4B).  In contrast, in Figure 1 the 3rd blood sample taken after 9 month is indicated as post treatment response sample. This applies to the other patients as well and does not seem to match at all?

The EV characterization in this study is somewhat minimal: Actin, Gapdh and Hspa8 are listed as proteins commonly present in small EVs. However, I would not rate them as specific marker proteins for small EVs as required by the MISEV guidelines. Instead, I would recommend to include other established, and more specific, marker proteins for small EVs (e.g. Syntenin, Alix, CD63) and also search for the presence of large EV marker proteins (e.g. ACTN4, MIC60 or RGAP1) to confirm that the preparations were indeed specifically enriched in small EVs and not large EVs. As MS results are always to be regarded with caution due to the high possibility of high positive/negative results, I would suggest to confirm marker expression by other methods, e.g. vesicle flow cytometry or immunoblot and also include common negative markers.

The observation of MAGE1 as a receptor for the association of IgG with EVs is currently only based on a MS result which is prone to false positive results. Can the authors validate this finding by additional methods and controls? E.g. pre-incubate PDAC cell line derived EVs with a MAGE1 blocking antibody followed by a spike-in of EVs to a plasma sample and MS analysis? Or validation of the interaction by co-immunoprecipitation and immunoblot?

Minor points:

Lipoproteins or Albumin are common contaminants in plasma EV preparations and also have been identified in the MS study conducted here (e.g. APOL1). In particular the identified IgG fragments are probably not exported on tumor EVs per se, but associate with tumor EVs once they are secreted into the circulation. Therefore, the “contamination” is not necessarily a bad thing as it led to the identification of the IgG fragments, but the authors should clearly mention in their Results/Discussion section that their proteomic results are not exclusively proteins expressed on EVs, but include soluble plasma proteins associated with EVs as well.

As NTA results are known to be influenced by many factors, can the authors add some more information for their measurements, e.g. sample diluent and dilution? With which laser was the NTA equipped? At which temperature were the measurements performed? How many positions and particles per frame were measured for each sample?

Line 236: The authors write “In fact, the functional analysis of proteins upregulated in EVs from PDAC patients as compared to healthy controls revealed enrichment in proteins associated with humoral immune response and complement activation, among others.” Was the analysis only performed on the significantly upregulated proteins? It would also be interesting to include the downregulated ones.

Line 244: The authors state that the majority of upregulated proteins in patients who did not respond to chemotherapy were IgG fragments. Can they give concrete numbers, how many of the 24 significantly upregulated proteins were IgG fragments?

Figure 4: For the indicated p values substitute , by .

Author Response

We thank the reviewer for the constrictive recommendations. Point-by-point responses are provided bellow.

Comments and Suggestions for Authors

In the manuscript “IgG+ extracellular vesicles measure therapeutic response in advanced pancreatic cancer” Couto and colleagues identify IgG+ EVs to be elevated in the plasma of pancreatic cancer patients and investigate their dynamics during chemotherapy. They demonstrate that IgG+ EVs levels correlate with treatment response and therefore represent a promising novel tool for PDAC therapy monitoring. The manuscript is clearly written and the clinical follow-up of the patients is documented in detail. As not many longitudinal studies have been performed for EVs as cancer biomarkers, this study is of high interest and has great potential for clinical translation. However, I would recommend to add some additional information and experiments to support the claims made by the authors: 

Major points:

While the authors give information on the PDAC patients used in the study, there is next to no information on the control group. 1 Therefore, I would ask the authors to include information on the cohort of healthy controls used for vesicle flow cytometry (age, gender). Moreover, to confirm the applicability of using IgG+ EVs as method for the detection of advanced PDAC, the authors should include benign controls, e.g. chronic pancreatitis, in particular as it is known that the described marker CA19.9 (and also IgG) is prone to variation under inflammatory conditions. Moreover, a ROC analysis would be helpful to judge the potential of IgG+ EVs as diagnostic marker. 

Considering the previous absence of information regarding the characteristics of the control group, we included on page 3 lines 116-119, information reinforcing the criteria used in the selection of the control group, gender distribution, and age of included individuals.

We agree with the Reviewer that as in CA19.9, we could not completely discard that inflammatory conditions could impact IgG+ EVs readings. However, although we show that levels of IgG+ EVs are higher in PDAC patients when compared to healthy control counterparts, the primary objective of the study was to evaluate IgG+EV as a marker of response in patients with a previous diagnosis of PDAC and not to use this biomarker as a diagnostic tool (where those inflammatory benign conditions have a higher impact). As such, when designing the study, we excluded (from both patients and control groups) individuals previously diagnosed with inflammatory conditions. Nonetheless, in order to address this potential interference, we used a validated clinical marker for inflammation (NLR), and we didn’t see significant differences between the groups in respect to inflammation status. To further clarify this issue, we now introduce in the discussion a comment on this point (lines 410-419).

We thank for the suggestion to perform a ROC analysis and integrate that analysis (Figure 4 and Supplementary Figure 3). We introduce the information of the ROC analysis in the methods (lines 235-237), and changes the results section accordingly on lines 317-321.

Currently, it is difficult to match the information of imaging response and course of IgG+ EV levels with the exact sample used for the evaluation of treatment response. For each patient the timepoints of sample collection have been indicated in Figure 1. However, not all of the measurements are shown in Figure 4B+C. Why? Moreover, the timelines shown in Figure 1 and Figure 4B+C do not match? The authors state that patients were allocated to the responder or non-responder group based on the timepoint of the best, or worst, imaging response. For example, for patient 15 the best imaging response would be for the first-line treatment in month 3 (Figure 4B).  In contrast, in Figure 1 the 3rd blood sample taken after 9 month is indicated as post treatment response sample. This applies to the other patients as well and does not seem to match at all? 

We thank the reviewer for pointing this inconsistence generated during the editing of the graphs. We corrected graphs 4B and C in order to match the timepoints displayed on the graphs with the month of collection. We now also include intermediate timepoints around imagiological evaluations to patients 15, 34 and 63. The missing points at month 2 and 3 in patients 8 and 15, respectively, were due to technical issues involving sample collection and storage prior to processing. As such, we now excluded these samples from the description in Figure 1.

The EV characterization in this study is somewhat minimal: Actin, Gapdh and Hspa8 are listed as proteins commonly present in small EVs. However, I would not rate them as specific marker proteins for small EVs as required by the MISEV guidelines. Instead, I would recommend to include other established, and more specific, marker proteins for small EVs (e.g. Syntenin, Alix, CD63) and also search for the presence of large EV marker proteins (e.g. ACTN4, MIC60 or RGAP1) to confirm that the preparations were indeed specifically enriched in small EVs and not large EVs. As MS results are always to be regarded with caution due to the high possibility of high positive/negative results, I would suggest to confirm marker expression by other methods, e.g. vesicle flow cytometry or immunoblot and also include common negative markers. 

We have re-evaluated our previous data, measuring the recommended markers. We now include the evaluation of the suggested markers by mass spectrometry in Supplementary Figure 2D, where we observed positivity to Syntenin and negativity to ACTN4, MIC60 and RGAP1. As suggested, it was further verified by Western blot analysis, which showed presence of CD9, CD81 and Alix and absence of GM130 and Calnexin. This result was included to Supplementary Figure 2E. The absence of CD63 in our analysis agrees with a comprehensive analysis of EV protein markers we performed across more than 100 different human tumor and non-tumor cell lines, which showed that only 40% of them are positive for CD63 (Hoshino, Ayuko et al. “Extracellular Vesicle and Particle Biomarkers Define Multiple Human Cancers.” Cell vol. 182,4 (2020): 1044-1061.e18. doi:10.1016/j.cell.2020.07.009).

The observation of MAGE1 as a receptor for the association of IgG with EVs is currently only based on a MS result which is prone to false positive results. Can the authors validate this finding by additional methods and controls? E.g. pre-incubate PDAC cell line derived EVs with a MAGE1 blocking antibody followed by a spike-in of EVs to a plasma sample and MS analysis? Or validation of the interaction by co-immunoprecipitation and immunoblot?

We agree that validation by additional methods and experimental designs would provide further support for the association of MAGEB1 with the binding of IgG to plasma EVs of PDAC patients. Due to the restricted number of plasma samples from PDAC patients fitting our inclusion criteria recruited in our single center study, the number of samples available to characterize the biological basis of IgG binding to PDAC EVs was limited. Furthermore, due to significant sample losses inherent in every step of the immunoprecipitation assay used to obtain surface proteins of EVs bound to IgG and necessary quality controls of protein integrity performed prior to sample analysis, the protein isolates from these patients had to be fully used in our protein MS analysis. Therefore, due to sample limitations, we unfortunately were not able to repeat our analysis by additional methods. In spite of that, as the interaction of IgG with MAGE1 and other proteins was verified in all of the 8 different PDAC patients studied in our MS analysis, we are confident that our results are unlikely to be false positive. We now acknowledge this limitation in our discussion (lines 446-448)

We agree that the suggested protein interaction experiment would have the potential to provide important verification of our results and conclusions. In not shown preliminary vesicle flow cytometry experiments, EVs from human PDAC cell lineages were mixed with EV-free plasma from 6 different PDAC patients. Interestingly, we found that no more than 2% of PDAC EVs were positive for IgG. This suggests, amongst others, that most of the plasma IgG specific to IgG-binding proteins should be already bound to plasma EVs and therefore unavailable for in vitro binding with PDAC EVs. Due to this important limitation, we didn’t further pursue in vitro interaction experiments such as the one suggested.

Minor points:

Lipoproteins or Albumin are common contaminants in plasma EV preparations and also have been identified in the MS study conducted here (e.g. APOL1). In particular the identified IgG fragments are probably not exported on tumor EVs per se, but associate with tumor EVs once they are secreted into the circulation. Therefore, the “contamination” is not necessarily a bad thing as it led to the identification of the IgG fragments, but the authors should clearly mention in their Results/Discussion section that their proteomic results are not exclusively proteins expressed on EVs, but include soluble plasma proteins associated with EVs as well. 

A sentence mentioning EVs post-secretion interactions and their role in the composition of EVs was now added to discussion (line 431-438).

As NTA results are known to be influenced by many factors, can the authors add some more information for their measurements, e.g. sample diluent and dilution? With which laser was the NTA equipped? At which temperature were the measurements performed? How many positions and particles per frame were measured for each sample?

This information was now added to item 2.2 of methods (lines 127-134).

Line 236: The authors write “In fact, the functional analysis of proteins upregulated in EVs from PDAC patients as compared to healthy controls revealed enrichment in proteins associated with humoral immune response and complement activation, among others.” Was the analysis only performed on the significantly upregulated proteins? It would also be interesting to include the downregulated ones. 

The analysis was performed on regulated genes (both up and down regulated). This information was included in the text (Line 257).

Line 244: The authors state that the majority of upregulated proteins in patients who did not respond to chemotherapy were IgG fragments. Can they give concrete numbers, how many of the 24 significantly upregulated proteins were IgG fragments?

We found that 16 of the 24 identified hits were IgG fragments. The text was updated with this information (Line 265).

Figure 4: For the indicated p values substitute , by .

The change was made.

Reviewer 2 Report

In the manuscript entitled “IgG+ Extracellular Vesicles measure therapeutic response in advanced pancreatic cancer” by Couto and colleagues, the authors characterize circulating plasma EVs from healthy and PDAC patients, focusing on differential protein expression between groups of patients. They show IgG fragments are significantly increased in EV preparations from PDAC patients, and validate this finding by flow cytometric techniques in a longitudinal study following several patients at various times of disease remission and progression. Overall, this study suggests a novel biomarker for tracking PDAC treatment response, and utilizes primary patient samples to draw their conclusions. However, several points should be addressed prior to publication:

1. Have the authors compared IgG+ EV levels across non-healthy, non-PDAC patient samples? Important controls would be patients with non-pancreatic cancer, or other inflammatory disorders (e.g. pancreatitis) to assess the specificity of this biomarker.

2. The authors use a biotinylation/co-IP technique to identify IgG-associated proteins on the surface of PDAC patient EVs, claiming that circulating IgG may bind neo-antigens on malignant cell-derived vesicles. However, in their mass spectrometry and flow cytometry-based assays, they do not necessarily demonstrate that the detectable IgG is surface-bound versus membrane embedded. Is IgG secreted into vesicles, and do these secreted levels correlate with treatment response?

3. Do IgG+ EV levels predict response to currently used immunotherapy (e.g. anti-PD1 therapy) or only chemotoxic therapy?

4. While trends of decreasing or increasing IgG+ EVs may be seen in responders vs nonresponders, respectively (Figure 4D-E), are these changes significant enough to predict response in an individual?

Author Response

We thank the reviewer for the constrictive recommendations. Point-by-point responses are provided bellow.

Comments and Suggestions for Authors

In the manuscript entitled “IgG+ Extracellular Vesicles measure therapeutic response in advanced pancreatic cancer” by Couto and colleagues, the authors characterize circulating plasma EVs from healthy and PDAC patients, focusing on differential protein expression between groups of patients. They show IgG fragments are significantly increased in EV preparations from PDAC patients, and validate this finding by flow cytometric techniques in a longitudinal study following several patients at various times of disease remission and progression. Overall, this study suggests a novel biomarker for tracking PDAC treatment response, and utilizes primary patient samples to draw their conclusions. However, several points should be addressed prior to publication:

  1. Have the authors compared IgG+ EV levels across non-healthy, non-PDAC patient samples? Important controls would be patients with non-pancreatic cancer, or other inflammatory disorders (e.g. pancreatitis) to assess the specificity of this biomarker.

We agree with the Reviewer that we could not completely discard that inflammatory conditions could impact IgG+ EVs readings. However, although we show that levels of IgG+ EVs are higher in PDAC patints when compared to healthy control counterparts, the primary objective of the study was to evaluate IgG+EV as a marker of response in patients with a previous diagnosis of PDAC and not to use this biomarker as a diagnostic tool (where those inflammatory benign conditions have a higher impact). As such, when designing the study, we excluded (from both patients and control groups) individuals previously diagnosed with inflammatory conditions. Nonetheless, in order to address this potential interference, we used a validated clinical marker for inflammation (NLR), and we didn’t see significant differences between the groups in respect to inflammation status. To further clarify this issue, we now introduce in the discussion a comment on this point (lines 410-419).

  1. The authors use a biotinylation/co-IP technique to identify IgG-associated proteins on the surface of PDAC patient EVs, claiming that circulating IgG may bind neo-antigens on malignant cell-derived vesicles. However, in their mass spectrometry and flow cytometry-based assays, they do not necessarily demonstrate that the detectable IgG is surface-bound versus membrane embedded. Is IgG secreted into vesicles, and do these secreted levels correlate with treatment response?

The MS analysis of proteins bound to IgG focused on EV surface proteins, as only these proteins were biotinylated during the immunoprecipitation protocol. Also, as vesicle flow cytometry studies were performed with intact EVs, all IgG+ signals derived from IgG bound at the external face of the EV surface. This agrees with the conventional mechanism of IgG binding to the surface of cells, which depends on the binding of IgG to a cell membrane protein on B cells, the Igα/Igβ heterodimer (CD79α/CD79β)2, 3. Therefore, based on that we consider that, as to CD79α/CD79β, the IgG anchorage on EVs relies on binding to surface proteins of EVs, such as MAGE B1. This information was now added to discussion (Lines 439-446)

  1. Do IgG+ EV levels predict response to currently used immunotherapy (e.g. anti-PD1 therapy) or only chemotoxic therapy?

Our patients’ cohort were submitted to current systemic treatments in accordance with decisions made in the multidisciplinary tumor board. The use of checkpoint inhibitors (like anti-PD1 therapy) in PDAC patients is only recommended for tumors with microsatellite instability, which represents less than 1% of the total PDAC population. Importantly, none of the patients included in our study belong to that group. Considering these points, such evaluation could not be performed in conventional cohorts of PDAC patients, such as ours. 

  1. While trends of decreasing or increasing IgG+ EVs may be seen in responders vs nonresponders, respectively (Figure 4D-E), are these changes significant enough to predict response in an individual?

Although our results indicate that IgG+ EVs correlate with imagiological respose of metastatic PDAC and, as consequence, can be used to measure response to chemotherapy, the absolute levels of this marker cannot predict the response to an specific treatment regimen.

Round 2

Reviewer 1 Report

I thank the authors for their great work and the careful and thorough revision of the manuscript. All my points have been adressed to my satisfaction and I recommend publication. 

Author Response

We thank the reviewer for the important suggestions and for recommending our manuscript for publication.  

Reviewer 2 Report

In attempts to address my first concern, the revised manuscript states that “levels of IgG+ EVs did not correlate with a validated clinical marker for inflammation (NLR), suggesting this biomarker is not affected by the inflammatory background of the patients.” Was a neutrophil:lymphocyte ratio correlated with IgG+ EV levels at each timepoint measurement for all patients? These data would be good to show, particularly if the authors are not going to include control patients with benign inflammatory conditions (e.g. pancreatitis). Correlation with additional markers of inflammation would be helpful, as NLR may be affected by chemotherapy treatment, and has been proposed as an independent prognostic marker in PDAC (Luo et al, PMID: 33269614; Xiang et al, PMID: 32510138).

In addition, despite the authors’ conclusion (lines 439-441), the initial mass spectrometry characterization of plasma EVs still does not necessarily prove that IgG is bound to the surface of EVs after biogenesis versus trafficked into EVs during biogenesis. In fact, the anti-IgG antibody used for flow cytometric analysis in this study recognizes the IgG F(ab')2. Doesn’t this suggest that the detected IgG is secreted/properly oriented within the EV membrane, rather than bound to an antigen at the surface? Limitations in these current conclusions should be addressed. Flow cytometric studies using an anti-IgG Fc antibody may also be considered.

Author Response

We would like to thank the reviewer for the very constructive comments that, in our opinion, certainly improved the revised version of our manuscript. We hope the new data and discussion satisfactorily  address the reviewer’s recommendations.

In attempts to address my first concern, the revised manuscript states that “levels of IgG+ EVs did not correlate with a validated clinical marker for inflammation (NLR), suggesting this biomarker is not affected by the inflammatory background of the patients.” Was a neutrophil:lymphocyte ratio correlated with IgG+ EV levels at each timepoint measurement for all patients? These data would be good to show, particularly if the authors are not going to include control patients with benign inflammatory conditions (e.g. pancreatitis). Correlation with additional markers of inflammation would be helpful, as NLR may be affected by chemotherapy treatment, and has been proposed as an independent prognostic marker in PDAC (Luo et al, PMID: 33269614; Xiang et al, PMID: 32510138).

A: We now include in supplementary figure S5, in lines 343-346 and line 420, correlation analysis of IgG+EVs vs NLR and vs an additional inflammatory marker (C-Reactive protein). We found that in both cases, IgG+EV levels did not correlate with inflammatory markers, reinforcing the hypothesis that this EV population does not seem to be affected by PDAC patients’ inflammatory background.

In addition, despite the authors’ conclusion (lines 439-441), the initial mass spectrometry characterization of plasma EVs still does not necessarily prove that IgG is bound to the surface of EVs after biogenesis versus trafficked into EVs during biogenesis. In fact, the anti-IgG antibody used for flow cytometric analysis in this study recognizes the IgG F(ab')2. Doesn’t this suggest that the detected IgG is secreted/properly oriented within the EV membrane, rather than bound to an antigen at the surface? Limitations in these current conclusions should be addressed. Flow cytometric studies using an anti-IgG Fc antibody may also be considered.

A: We now include discussion on previous studies showing post-secretion IgG interaction with circulating EVs (lines 435-438). We also acknowledge  recent reports on the production and secretion of IgG by tumor cells (including PDAC – lines 456-463), which in agreement with the reviewer’s recommendation could correspond (at least partially) to IgG+EVs produced directly by tumor cells.

Round 3

Reviewer 2 Report

Comments have been adequately addressed.